# Effects of Chinese Yam Polysaccharide on Intramuscular Fat and Fatty Acid Composition in Breast and Thigh Muscles of Broilers

**DOI:** 10.3390/foods12071479

**Published:** 2023-03-31

**Authors:** Liping Guo, Yadi Chang, Zhe Sun, Jiahua Deng, Yan Jin, Mingyan Shi, Jinzhou Zhang, Zhiguo Miao

**Affiliations:** 1School of Food Science and Technology, Henan Institute of Science and Technology, Xinxiang 453003, China; 2College of Animal Science and Veterinary Medicine, Henan Institute of Science and Technology, Xinxiang 453003, China; 3College of Life Science, Luoyang Normal University, Jiqing Road, Luoyang 471022, China

**Keywords:** Chinese yam polysaccharide, intramuscular fat, fatty acids

## Abstract

The purpose of this study is to evaluate the influences of Chinese yam polysaccharide (CYP) dietary supplementation on the composition of intramuscular fat (IMF) and fatty acids (FA) in thigh and breast muscles of broilers. Three hundred and sixty healthy one-day-old broilers (the breed of Crossbred chicken is named 817) with gender-balanced and similar body weight (39 ± 1 g) were randomly allocated into four groups (control, CYP1, CYP2, and CYP3 groups). Broilers in the control group were only fed a basal diet, and broilers in CYP1 group were fed the same diets further supplemented with 250 mg/kg CYP, the CYP2 group was fed the same diets further supplemented with 500 mg/kg CYP, and the CYP3 group was fed the same diets further supplemented with 1000 mg/kg CYP, respectively. Each group consisted of three replicates and each replicate consisted of 30 birds. The feeding days were 48 days. The results observed that the CYP2 group (500 mg/kg) can up-regulate the mRNA expression levels of *β-catenin* in thigh muscle compared to the control group. At the same time, all CYP groups (CYP1, CYP2, and CYP3 groups) can up-regulate mRNA expression of *Wnt1* and *β-catenin* in breast muscle, while mRNA expression of *PPARγ* and *C/EBPα* in breast and thigh muscles could be down-regulated (*p* < 0.05). In summary, 500 mg/kg of CYP dietary supplementation can reduce IMF content and improve the FAs composition, enhancing the nutritional value of chicken meat.

## 1. Introduction

The maintenance of a balanced diet throughout life for health and well-being is very important. Over the last few decades, due to changes in living standards and nutritional ideologies of some people, the development of the levels of global production and consumption of meat and meat products has become rapid. Therefore, it is very important to address different aspects of meat and meat products, especially regarding nutrition and safety. Fat is not only a vital nutrient in meat, but also an important indicator for evaluating meat quality. Excessive fat deposition reduces feed efficiency, the percentage of lean meat, and deteriorates meat quality, and can even lead to fatty liver disease, which can cause serious health problems [1]. It has been established in epidemiological and experimental studies that high fat intake contributes to obesity and is associated with inflammation of the adipose tissue, necrotic cell death in adipocytes, and stress of the endoplasmic reticulum of adipocyte [2,3]. In the world, there are over 1.1 billion overweight adults and about 115 million people in low-income and middle-income groups are experiencing obesity-related problems, as estimated by World Health Organization (WHO) [4]. Intramuscular fat (IMF), which refers to the amount of fat that is deposited between and within muscle fibers, is a type of fat that includes endomysium, muscle bundle membrane, and epimysium [5]. The difference from other adipose tissue is that the IMF has its own characteristics, and its main lipids are triglyceride (TG), cholesterol (CHO), and large amounts of phospholipids (PL) [6]. Furthermore, the IMF consists of a variety of polyunsaturated fatty acids (PUFAs) containing linoleic acid (18:2 *n*-6), α-linolenic acid (18:3 *n*-3), arachidonic acid (C20:4 *n*-6), etc., which is fortifies human health [7]. IMF deposition mainly depends on the differentiation, maturation, and proliferation of intramuscular preadipocytes [8]. IMF deposition is influenced by various factors, including heredity, environment, feeding management, and trophic regulation. In recent years, many nutritionists have found that the fatty acid composition and fat content of muscle can be influenced by feed additives, which can improve meat quality [9,10].

In broiler breeding, natural polysaccharides are widely used to enhance the performance and meat quality that can be used as functional feed additives to alternative antibiotics [11,12]. Yam (*Dioscorea* spp.), which is a plant of medicinal and edible value, contains phenols, polyphenols, peptides, alkaloids, terpenoids, steroids, and essential oils and other bioactive ingredients [13]. Compared to other tropical tuber crops, it contains high levels of protein, dietary minerals, essential amino acids, and many other nutrients that make it a good essential dietary supplement [14,15]. As reported in pharmacological research studies, it possesses antimutagenic, anti-microbial, anti-fungal, immunomodulatory, and hypoglycemic properties [16]. Chen et al. [17] found that yam can enhance the metabolism of protein and fat, promote amino acid utilization, and reduce fat accumulation, thus improving the nutritional value of chicken. Kusano et al. [18] found that the triglyceride contents of the viscera, abdomen, and quadriceps of male Wistar rats can be reduced by Japanese yam dietary supplementation. McKoy et al. [19] also found that Jamaican bitter yam inhibited the deposition of large fat in the liver by a high cholesterol diet and concluded that long-term supplementation of Jamaican bitter yam may induce some changes in kidney and liver function. Furthermore, the Chinese yam peel can be used as an additive in feed to improve the microbiome of fish [20]. Chinese Yam polysaccharides (CYP), which are considered to be its main active component, include homopolysaccharides, heteropolysaccharides, and glycoproteins [21]. In recent years, CYP have become an important research direction in yam due to their biological activities with a wide range of pharmacological effects. Nishimura et al. [22] found that Chinese yam starch has the ability to promote cecal fermentation and reduce plasma non-HDL cholesterol and triglyceride concentrations in rats. This finding could be attributed to the fact that the dry tuber weight has a resistant starch of 50% [22,23,24,25]. Yu et al. [26] found that nano-yam polysaccharide can reduce serum TC, TG, and LDL-C content of diabetic rats fed alloxan and improve their symptoms. Zhao et al. [27] found that yam polysaccharide, as a dietary additive, can improve immunity, intestinal morphology, and digestive enzyme activity of weaned piglets. In addition, our previous related studies also show that CYP was beneficial in enhancing immune function and antioxidant capacity of broilers [28,29]. This research was conducted to investigate the effects of CYP on intramuscular fat deposition and fatty acid composition of broilers, and to provide a basis for the scientific application of CYP.

## 2. Materials and Methods

### 2.1. Experimental Design and Diets

The CYP required in this study came from Shaanxi Hannah Biotechnology Co., Ltd. (Xi’an, China), with 80-mesh granularity, a dry loss weight of not more than 5%, and polysaccharide content of 30% (monosaccharide types include 99.48% glucose and 0.52% galactose). In this study, all treatment protocols for experimental birds were approved by the Animal Care and Use Committee of Henan Institute of Science and Technology (No. 2020HIST018, Xinxiang, China). A total of 360 healthy broilers (the breed of Crossbred chicken is 817) with a mean weight of 39 ± 1 g were randomly divided into four groups, namely one control group (basic diet only) and three test groups (basal diet was supplemented with 250, 500, and 1000 mg/kg CYP, respectively), 30 birds in each group. Animal testing was divided into two stages (128 d and 2948 d). The basic diet was prepared according to the nutritional requirements of broilers set by the National Research Council [30], and the ingredients of the basal diet are shown in Table 1. The birds were kept in cages, and the feeding management followed the routine production and immunization process of Henan Fengyuan Poultry Co., Ltd. (Xinxiang, China). The cages and rooms used for animal raising were fully cleaned and disinfected prior to the start of the experiment. Broilers had free access to water and feed, and for the first 3 days, all birds were exposed to incandescent light for 24 h at 32 °C and 80% humidity, then cooled down regularly, to 26 °C on day 21.

### 2.2. Sample Collection

Two birds of similar weight were randomly taken from each replicate of each treatment group (total 24 birds; mean weights were as follows: control group, 1.42 kg; CYP1 group, 1.60 kg; CYP2 group, 1.68 kg; CYP3 group, 1.58 kg), dissected immediately after euthanasia, and the muscle tissues of the breast and thigh were collected in 1.5 mL centrifuge tubes and then quickly placed in liquid nitrogen. When the samples were fully collected, they were transferred to an ultra-cryogenic refrigerator at −80 °C (Golden Youning Instruments Co., Ltd., Zhengzhou, China) for measuring fatty acid composition and quantitative real-time polymerase chain reaction (qRT-PCR) tests.

### 2.3. Intramuscular Fat Content and Fatty Acid Composition Analysis

The IMF (dry sample) contents of thigh and breast muscles were determined by Soxhlet extraction after skin and significant fat were removed [31]. The results were expressed as the percentage of the weight of dried meat samples after extraction to that of dried meat samples in advance. Fatty acid composition was determined according to the method of Sukhija et al. after skin and significant fat were removed [32]. Lipids were extracted from freeze-dried breast and thigh muscle tissue by one-step method for methyl ester, and the supernatant was filtered and analyzed by gas chromatograph (Agilent-7890, Agilent Technologies Inc., Santa Clara, CA, USA). The same determination was made with the standard sample. The concentration of each fatty acid was obtained by standard curve, and the fatty acid composition was measured by area normalization method, and the fatty acid was expressed in %.

### 2.4. Nutritional Indexes

The fatty acids (FAs) were divided into saturated fatty acids (SFAs), monounsaturated fatty acids (MUFAs), PUFAs, DHA + EPA (docosahexaenoic acid, C22:6 *n*-3; eicosapentaenoic acid, C20:5 *n*-3), and the ratio of *n*-6: *n*-3 and PUFAs: SFAs were calculated. In addition, the fatty acid unsaturation index (UI), peroxidation trend index (PI), nutrition value index (NVI) and health-promoting index (HPI), index of thrombogenicity (IT) and index of atherogenicity (IA), and hypocholesterolemic/hypercholesterolemic (HH) ratio were calculated according to Logue et al. [33], Witting et al. [34], Chen et al. [35], Ulbricht and Southgate [36], and Santos-Silva et al. [37], respectively. The calculation formula for each index is as follows:
PUFA: SFA = PUFAs/SFAsDHA + EPA = (C22:6n-3 %) + (% C20:5n-3)UI = (% monoenoics) × 1 + (% dienoics) × 2 + (% trienoics) × 3 + (% tetraenoics) × 4 + (% pentaenoics) × 5 + (% hexaenoics) × 6PI = (0.025 × percentage of monoenoic acid) + (1 × percentage of dienoic acid) + (2 × percentage of trienoic acid) + (4 × percentage of tetraenoic acid) + (6 × percentage of pentaenoic acid) + (8 × percentage of hexaenoic acid).NVI = (C18:0 + C18:1n9)/(C16:0)IA = (4 × C14:0 + C16:0)/(Σ MUFA + Σ PUFA)IT = (C14:0 + C16:0 + C18:0)/[(0.5 × MUFA) + (0.5 × Σn-6 PUFA) + (3 × Σn-3 PUFA) + (Σn-3 PUFA/Σn-6 PUFA)]HH ratio = (C18:1 + Σ PUFA)/(C14:0 + C16:0)HPI = (Σ UFA)/(4 × C14:0 + C16:0)

### 2.5. qRT-PCR

Primer sequences were designed based on known sequences stored in GenBank (Table 2). Total RNA was extracted from thigh and breast muscles with TRIzol reagent (Takara Bio Inc., Tokyo, Japan) and a spectrophotometer (IMPLEN, Westlake Village, CA, USA) was used to measure the purity and concentration of extracted RNA at optical density 1.8 ≤ 260/280 ≤ 2.2. Complementary DNA (cDNA) was synthetized using the PrimeScript RT Reagent Kit (Takara Bio Inc., Tokyo, Japan). The expression levels of the genes involved in IMF were measured by qRT-PCR. Bio-Rad SYBR green RT-PCR kit was used for qRT-PCR (Hercules, CA, USA) and the ViiA 7 real-time PCR system. With *β-actin* as internal reference, the 2^−ΔΔCT^ method was used to calculate the expression levels of mRNA.

### 2.6. Statistical Analysis

SPSS 26.0 (SPSS Inc., Chicago, IL, USA) was used to perform the statistical analysis of data, and the statistical significance was measured using the Duncan multiple range test using one-way ANOVA. Data were visualized using GraphPad Prism 6 software (GraphPad, San Diego, CA, USA). Data were considered to be different at *p* < 0.05, and the results are presented as mean ± SEM.

## 3. Results

### 3.1. Intramuscular Fat

Compared to the control group, adding 500 mg/kg CYP to the diet can reduce the IMF content in breast muscle of broilers (*p* < 0.05), and dietary supplementation of 250 mg/kg and 500 mg CYP can reduce the IMF content in thigh muscle of broilers (*p* < 0.05), and the lowest values were found in the CYP2 group (Table 3).

### 3.2. Intramuscular Fat Related Gene Expression

The results of dietary CYP supplementation on the expression of adipose-related genes in the thigh muscle tissue of broilers are shown in Figure 1. Compared to the control group, the CYP2 group had higher *β-catenin* mRNA expression level (*p* < 0.05), all experimental groups had lower *PPARγ* and *CEBP/α* mRNA expression levels (*p* < 0.05), and there was no significant effect on the Wnt1 mRNA expression level of all groups (*p* > 0.05). Moreover, the *PPARγ* mRNA expression level in the CYP2 group was lower than in the CYP3 group (*p* < 0.05).

In breast muscle tissue, it can be drawn from Figure 2 that all experimental groups had a higher level of *Wnt1* and *β-catenin* mRNA expression (*p* < 0.05), while they had lower *PPARγ* and *CEBP/α* mRNA expression levels (*p* < 0.05) for broilers compared to the control group. Meanwhile, the mRNA expression level of *β-catenin* in the CYP2 group was higher than that in the CYP3 group but lower than that in the CYP1 group (*p* < 0.05), the mRNA expression level of *PPARγ* in the CYP1 group was higher than that in the CYP1 and CYP3 groups (*p* < 0.05), and there were no significant differences in mRNA expression levels of *Wnt1* and *CEBP/α* among experimental groups (*p* > 0.05).

### 3.3. Fatty Acid Composition

For breast muscle, the experimental data obtained are shown in Table 4, compared to the control group, all experimental groups had lower SFA concentrations (*p* < 0.05), and the CYP2 group was significantly lower than the CYP1 and CYP3 groups (*p* < 0.05). The concentrations of C14:0, C18:0, and C20:0 in the CYP2 group were significantly lower than those of the control group (*p* < 0.05). On the contrary, the CYP2 group had lower C14:0 and C18:0 concentrations compared to the CYP1 group (*p* < 0.05) and lower C14:0 and C20:0 concentrations compared to the CYP3 group (*p* < 0.05). Table 5 shows the result of thigh muscle tissue, compared to the control and CYP3 groups, broilers in CYP1 and CYP2 groups had lower SFA concentrations (*p* < 0.05). The C14:0, C16:0, and C20:0 concentrations of the CYP2 group were significantly lower than those of the control group (*p* < 0.05). All CYP groups had lower C18:0 concentrations compared with the control group (*p* < 0.05). In addition, the CYP2 group had lower C14:0 concentrations compared to the CYP1 and CYP3 groups (*p* < 0.05) and the CYP1 and CYP2 groups had lower C16:0 and C20:0 concentrations compared to the CYP3 group (*p* < 0.05).

In breast muscle, the experimental data obtained are shown in Table 4, there were no significant differences in MUFA concentrations among all groups (*p* > 0.05). However, the concentrations of C14:1, C20:1, C22:1, and C24:1 of the CYP2 group were significantly lower than those of the control group (*p* < 0.05), and the CYP2 group had lower concentrations of C14:1 and C22:1 compared to the CYP1 group and lower C22:1 concentration compared to the CYP3 group (*p* < 0.05). However, the CYP1 group had lower C16:1 concentration compared to other groups (*p* < 0.05). In addition, the C18:1 concentration was not significantly different among the control and all experimental groups. Table 5 shows the result of thigh muscle tissue, CYP2 and CYP3 groups had lower MUFA concentrations than the control group, and the concentration of MUFA in the CYP3 group was significantly lower than that in the CYP1 group (*p* < 0.05). The C14:1, C16:1, C20:1, and C24:1 concentrations of the CYP2 group were significantly lower than those of the control group (*p* < 0.05). In addition, the CYP2 group had lower C14:1 concentration compared to the CYP3 group (*p* < 0.05), and there were lower C16:1 concentrations in the CYP1 and CYP2 groups compared to the CYP3 group (*p* < 0.05), and the CYP2 and CYP3 groups had lowerC20:1 and C24:1 concentrations compared to the CYP2 group (*p* < 0.05). However, the C18:1 concentrations of the CYP1 and CYP2 groups were significantly higher than those of the control and CYP3 groups (*p* < 0.05).

In breast muscle, the experimental data obtained are shown in Table 4. Compared to the control group, broilers in the CYP2 group had higher PUFA concentrations (*p* < 0.05). Additionally, the PUFA concentrations in the CYP2 group were significantly higher than those of the CYP1 and CYP3 groups (*p* < 0.05). The C18:2 *n*-6, C18:3 *n*-3, and C22:6 *n*-3 concentrations of the CYP2 group were significantly higher than those of the control group (*p* < 0.05), at the same time, the C18:2 *n*-6 and C18:3 *n*-3 concentrations of the CYP2 group were significantly higher than those of the CYP1 and CYP3 groups (*p* < 0.05), whereas, the C20:4 *n*-6 concentrations of the CYP2 group were significantly lower than those of the control group (*p* < 0.05). In addition, the C20:5 *n*-3 concentrations of the CYP3 group were significantly higher than those of the control and CYP2 groups (*p* < 0.05). Additionally, there was no significant difference among groups in the concentrations of the C18:3 *n*-6, C20:2 *n*-6, and C20:3 *n*-6 (*p* > 0.05).

Table 5 shows the result of thigh muscle tissue. In the thigh muscle, compared to the control group, the broilers in the CYP2 and CYP3 groups had higher PUFA concentrations (*p* < 0.05), and the PUFA concentrations in the CYP2 group were significantly higher than those of the CYP1 and CYP3 groups (*p* < 0.05). The C18:2 *n*-6 concentrations of the CYP2 group were significantly higher than those of the control and CYP1 groups (*p* < 0.05). At the same time, the C20:5 *n*-3 concentrations of the CYP group were significantly higher than those of the control group (*p* < 0.05). The C22:6 *n*-3 concentrations of the CYP1 and CYP2 groups were significantly higher than those of the control and CYP3 groups (*p* < 0.05), whereas, the C18:3 *n*-3 concentrations of the CYP1 group were significantly lower than those of the other groups (*p* < 0.05), and the CYP3 group had higher C20:4 *n*-6 concentrations compared to the other groups (*p* < 0.05). Additionally, there were no significant differences among all groups in the concentrations of the C18:3 *n*-6, C20:2 *n*-6, and C20:3 *n*-6 (*p* > 0.05).

### 3.4. Nutritional Indicators of Fatty Acid Composition

As indicated in Table 6, in breast muscle, the *n*-6: *n*-3 in the CYP groups were significantly lower than the control group. The IT in the CYP2 group were significantly lower than the other groups (*p* < 0.05), and the IT in the CYP1 group were also significantly lower than the control group (*p* < 0.05). The DHA + EPA and PI in the CYP groups were significantly higher than the control group (*p* < 0.05) and there were no significant differences among them (*p* > 0.05). The total PUFA: SFA in the CYP2 group was significantly higher than the control, CYP1, and CYP3 groups; and there were no significant differences for UI, NVI, IA, HHR, and HPI among all groups (*p* > 0.05).

Nutritional indices for assessing fatty acids of thigh muscle are presented in Table 7. The PUFA: SFA and PI in the CYP2 group had the highest values, and the CYP1 and CYP3 groups had lower PUFA/SFA and PI than the CYP2 group, but higher than the control group (*p* < 0.05). The DHA + EPA, NVI, HHR, and HPI in the CYP1 and CYP2 groups were significantly higher than in the control and CYP3 groups (*p* < 0.05), and the DHA + EPA in the CYP3 group was also significantly higher than in the control group (*p* < 0.05). The *n*-6/*n*-3 in the CYP2 group was significantly lower than in the other groups (*p* < 0.05). The IA and IT in the CYP1 and CYP2 groups were significantly lower than in the control and CYP3 groups (*p* < 0.05).

## 4. Discussion

IMF contributes a key factor to meat quality [38]. This process is regulated by lipid-forming transcription factors such as the *C/EBPα* and *PPARγ*. *PPARγ*, one of the nuclear hormone receptors superfamilies, is a ligand-activated transcription factor. Previous studies have shown that *PPARγ* is a major regulator of adipogenesis, controlling adipocyte differentiation, proliferation, and lipid deposition [39]. In addition, Supanon et al. [40] found a moderately positive correlation between intramuscular fat and PPARγ expression in breast and thigh tissues of chickens. *C/EBPα*, a member of the C/EBP family, plays an important role in the differentiation of adipocytes. Previous studies have shown that adipocyte differentiation can be blocked by inhibition of *C/EBPα* mRNA expression, and it is also associated with lipid accumulation in intramuscular preadipocytes [41,42]. Wnt signal transduction has shown the ability to inhibit adipogenesis by blocking the induction of *PPARγ* and *C/EBPα* [43]. Its scope is the stability of β-catenin protein, which performs a dual role in the formation of intercellular junctions and transcriptional regulation. Wnt1 is the first member of the Wnt family to be identified as highly conserved. Moldes et al. [44] showed that Wntl inhibits adipocyte differentiation by reducing the expression of *C/EBPα* and *PPARγ*. Cai et al. [45] believed in their study that reducing the level of mRNA and protein in Wnt1 can affect the differentiation of muscle cells, and the expression level of Wnt1 has an important effect on promoting muscle synthesis and reducing fat deposition. The effect of plant extracts on lipid deposition by regulating the expression level of lipid metabolism-related genes has been demonstrated in previous studies [46,47]. At the same time, according to the study of Xu et al. [48], adding different dosages of plant sterols to feed can up-regulate the expression level of *Wnt1* mRNA in liver and back muscle of pigs, thereby improving the lean meat rate. In our study, the CYP2 group can reduce the IMF content in breast muscle of broilers, and CYP1 and CYP2 groups can reduce the IMF content in thigh muscle of broilers. This finding could be attributed to the fact that the CYP groups had lower mRNA expression of *PPARγ* and *C/EBPα* in thigh and breast muscles compared to controls, and the CYP2 group had higher mRNA expression of *β-catenin* in the thigh muscle and the CYP groups had higher mRNA expression of *wnt1* and *β-catenin* in the breast muscle compared to the control group, whereas, IMF content was increased in the CYP3 group. In our previous study, the immune capacity of broilers was decreased when CYP content was increased to 1000 mg/kg [28]. The increase in IMF content may be related to the metabolic burden caused by high dose of yam polysaccharide on the body, which may reduce the immune performance of broilers. Further research is needed to prove the relevant mechanism. Therefore, the addition of 500 mg/Kg CYP to the diet inhibits fat deposition within the muscle, suggesting that the product may be more popular with those who prefer a high percentage of lean meat.

FAs are carboxylate compounds composed of carbon, hydrogen, and oxygen. For broiler meat, the type and content of FAs are important factors that affect its nutritional value and taste, and probably influences their storage or further processing. Excessive consumption of SFAs has been demonstrated to result in elevated cholesterol levels, thereby increasing the odds of cardiovascular disease [49]. The *n*-6 and *n*-3 fatty acids are long chain polyunsaturated fatty acids, which are crucial components of the human diet. The *n*-3 FAs include α-linolenic acid (ALA, C18:3 *n*-3), EPA, DHA, etc., which are helpful to the brain and cardiovascular system [50]. The *n*-6 FAs include linoleic acid (LA, C18:2 *n*-6), arachidonic acid (ARA, C20:4 *n*-6), etc., and as it has been found LA has positive effects on reducing cholesterol concentration of serum and preventing cardiovascular disease [51]. Furthermore, EPA and DHA have anti-inflammatory or anti-inflammatory properties, whereas SFAs are the opposite [52,53]. Our study found that the SFAs of breast muscle in the control group were higher than the CYP groups and the SFAs of thigh muscle were lower in the CYP1 group and CYP2 group compared to the control group in broilers. Moreover, our study showed that the concentrations of *n*-3 PUFAs (C18:3 *n*-3, C22:6 *n*-3) in the breast muscle and *n*-3 PUFAs (C20:5 *n*-3, C22:6 *n*-3) in the thigh muscle were significantly higher in the CYP2 group compared to the control group (*p* < 0.05). Furthermore, the concentrations of C18:2 *n*-6 in the breast and thigh muscles were significantly higher in the CYP2 group compared to the control group (*p* < 0.05), which may indicate that the yam polysaccharide is an effective substance to improve FAs profile of broilers. Our findings were similar to that of dietary chitosan supplementation in the diets of Huoyan geese [11]. These results suggested that adding 500 mg/Kg CYP to the diet can enhance broiler meat nutritional value by improving fatty acids composition in breast and thigh muscles.

PUFA/SFA is commonly used as an index to evaluate the effect of food on cardiovascular health (CVH). All PUFAs were assumed to reduce LDL-C and serum cholesterol, while all SFAs can increase serum cholesterol. Thus, PUFA/SFA is an immediate index; its value is negatively correlated with health status. Nowadays, n6/n3 has become a method used by many researchers to evaluate the nutritional quality of foods. Additionally, a lower *n*-6/*n*-3 ratio is associated with a lower incidence of many chronic diseases in Western societies and developing countries [54]. UI indicates the degree of unsaturation of fatty acid and is useful to establish the oxidative stability of human food or livestock feed and to identify some strategies for oxidative protection [55]. The peroxidizability index provides a representative correlation between the FA composition in a tissue and its oxidation sensitivity and signifies the quality of the meat process. Furthermore, the PI can be used to evaluate the stability of the PUFAs contained in foods and to keep them from possible oxidation processes. In addition, higher PI value has protective potential to reduce the incidence of coronary artery disease. NVI can be used to describe the potential health effects of different types of lipids, and it is positively correlated with FAs quality [56]. The IA and IT were proposed by Ulbritcht et al. [36] to describe the atherogenic and thrombogenic potential of FAs, respectively. LDL-C and total cholesterol levels in human blood plasma have been reported to be reduced by consuming foods or products with lower IA content [57]. IT indicates not only the tendency of FAs to form clots in blood vessels but also the correlation between the prothrombogenic FAs (C12:0, C14:0, and C16:0) and antithrombogenic FAs (MUFAs, *n*-3 and *n*-6 families) [36]. The HHR shows the effect of FAs on blood cholesterol level, and the HHR value is positively correlated with health level. It has been reported that a relatively high HHR with low IA and IT contributes to a lower incidence of coronary heart disease [58]. The HPI is the opposite of IA, the higher the value, the better the nutritional quality of the product under study. Bobe et al. [59] found that dietary supplementation with fish oil or baked soy could improve HPI in dairy milk, but the difference was not significant. Omri et al. [60] found that feeding linseed-supplemented diets did not affect the HHR and the IA of egg yolks, but significantly reduced the *n*-6/*n*-3 and IT. Mir et al. [61] found that dietary flaxseed supplementation significantly increased the PUFA:SFA ratio of broiler meat, thereby reducing IA and IT. A growing body of studies have found that natural plant extracts can improve the nutritional value of fatty acids in animal products. In our study, there is higher PUFA: SFA and PI, and lower *n*-6: *n*-3 and TI in the breast muscle of the CYP2 group compared to the control group. In thigh muscle, the PUFA: SFA, PI, NVI, HHR, and HPI in the CYP2 group were higher than in the control group, and *n*-6: *n*-3, AI, and TI had the contrary result. Therefore, the consumption of broiler products supplemented with 500 mg/kg CYP in the diet is beneficial for human health.

## 5. Conclusions

Dietary supplementation of 500 mg/Kg yam polysaccharide can inhibit IMF deposition in breast and thigh muscles of broilers and improve fatty acid composition of broilers, thus making it more beneficial to human health.

## Figures and Tables

**Figure 1 foods-12-01479-f001:**
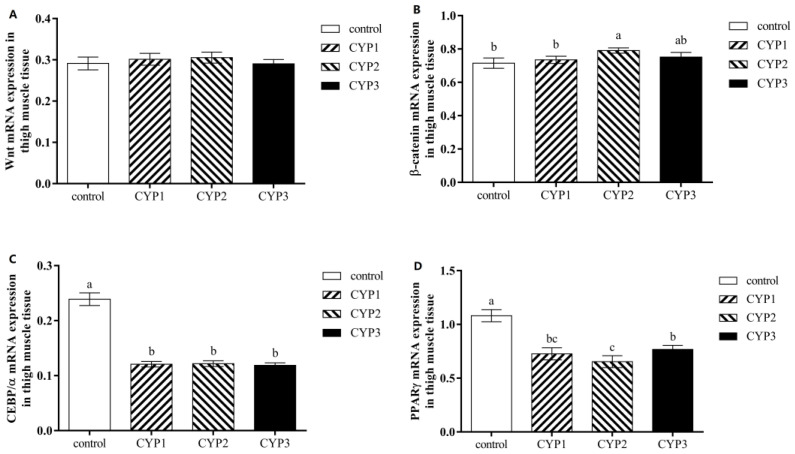
The *Wnt1* (**A**), *β-catenin* (**B**), *CEBPα* (**C**), and *PPARγ* (**D**) mRNA expressions in thigh muscle of broilers. ^a,b,c^ The difference between the small letters indicates a significant difference (*p* < 0.05). Control, basic diet; CYP1, CYP2, and CYP3 represent the basal diet supplemented with 250, 500, and 1000 CYP, respectively. *PPARγ*, peroxisome proliferation-activated receptor γ; *C/EBPα*, CCAAT/enhancer binding protein α.

**Figure 2 foods-12-01479-f002:**
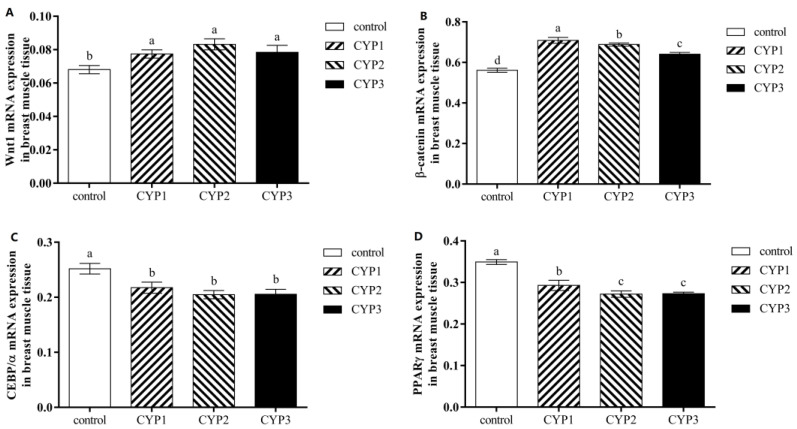
The *Wnt1* (**A**), *β-catenin* (**B**), *CEBPα* (**C**), and *PPARγ* (**D**) mRNA expressions in breast muscle of broilers. Control, CYP1, CYP2, CYP3, *PPARγ*, and *C/EBPα* same for Figure 1. ^a,b,c^ The difference between the small letters indicates a significant difference (*p* < 0.05).

**Table 1 foods-12-01479-t001:** Composition and nutrition level of basic diet.

Ingredients	Content	
	1–28 Days	29–48 Days
Corn (%)	60.00	63.50
Soybean meal (%)	32.00	29.00
Wheat bran (%)	1.00	
Soybean oil (%)	1.00	2.00
Fish meal (%)	2.00	1.60
CaHPO4 (%)	1.30	1.30
Lime stone (%)	1.40	1.30
NaCl (%)	0.30	0.30
Premix ^1^ (%)	1.00	1.00
Total (%)	100.00	100.00
Nutrient levels		
ME/(MJ·kg^−1^) ^2^	12.13	12.55
CP (%)	21.00	20.00
Ca (%)	1.00	0.90
TP (%)	0.65	0.60
AP (%)	0.45	0.35
Lys (%)	0.50	0.38
Met (%)	1.10	1.00

^1^ The premix amounts of VA, VD_3,_ VE, VK_3,_ VB_6,_ VB_1,_ D-pantothenic acid, folic acid, biotin, Fe, Cu, Zn, Mn, and Se per kg were 3000 IU, 500 IU, 10 IU, 0.5 mg, 3.5 mg, 3.8 mg, 10 mg, 0.5 mg, 0.15 mg, 80 mg, 8 mg, 75 mg, 60 mg, and 0.15 mg per kg of diets, respectively. ^2^ Other than ME (calculated value) were measured values.

**Table 2 foods-12-01479-t002:** Primer sequences for qRT-PCR analysis.

Genes (Accession)	Primer Sequence	Length (bp)
*PPARγ* (NM_001001460)	F:5′-CGAATGCCACAAGCGGAGAAGG-3′ R:5′-CACTGCCTCCACAGAGCGAAAC-3′	330
*C/EBPα* (NM_001031459)	F:5′-GCCAACTTCTACGAGGTCGATTCC-3′ R:5′-TTGTGCTTCTCCTGCTGCTTGC-3′	260
*Wnt1* (NM_001396681)	F:5′-GGCTCTTCGGGAGGGAATTTGTG-3′ R:5′-TGCCTTTGTTGCCGTAGATGACC-3′	255
*β-catenin* (U82964)	F:5′-GCTATTGTTGAGGCTGGTGGGATG-3′ R:5′-GCTTCCTGATGTCTGCTGGTGAG-3′	368
*β-actin* (L08165)	F:5′-CATTGAACACGGTATTGTCACCAACTG-3′ R:5′-GTAACACCATCACCAGAGTCCATCAC-3′	270

**Table 3 foods-12-01479-t003:** Effects of CYP on IMF content from breast and thigh muscles in broilers.

Item	CYP Level (mg/kg) ^1^				SEM ^2^	*p*-Value
	Control	CYP1	CYP2	CYP3		
Breast muscle fat ratio (%)	6.25 ^a^	6.09 ^a^	5.45 ^b^	5.96 ^ab^	0.22	0.032
Thigh muscle fat ratio (%)	24.41 ^a^	22.07 ^b^	21.45 ^b^	23.55 ^a^	0.59	0.004

^1^ Control, basic diet; CYP1, CYP2, and CYP3 represent the basal diet supplemented with 250, 500, and 1000 mg/kg CYP, respectively. ^2^ SEM, standard error of the mean. ^a,b^ Different lowercase letters show significant differences on the same line (*p* < 0.05).

**Table 4 foods-12-01479-t004:** Effects of CYP on fatty acids composition from breast muscle in broilers.

Item	CYP Level (mg/kg) ^1^				SEM ^2^	*p*-Value
	Control	CYP1	CYP2	CYP3		
SFA						
C14:0 (%)	0.46 ^a^	0.40 ^b^	0.35 ^c^	0.47 ^a^	0.021	0.001
C16:0 (%)	25.71	24.85	24.87	25.44	0.43	0.098
C17:0 (%)	4.49	4.32	4.46	4.54	0.24	0.798
C18:0 (%)	10.46 ^a^	10.40 ^a^	9.37 ^b^	9.80 ^ab^	0.24	0.017
C20:0 (%)	0.041 ^a^	0.029 ^b^	0.025 ^b^	0.038 ^a^	0.0021	0.001
MUFA						
C14:1(%)	0.052 ^a^	0.047 ^a^	0.035 ^b^	0.034 ^b^	0.0035	0.002
C16:1 (%)	1.36 ^a^	1.19 ^b^	1.41 ^a^	1.39 ^a^	0.072	0.05
C18:1 (%)	30.03	30.32	30.79	28.88	0.92	0.273
C20:1 (%)	0.41 ^a^	0.30 ^b^	0.25 ^b^	0.27 ^b^	0.03	0.003
C22:1 (%)	0.10 ^a^	0.10 ^a^	0.059 ^b^	0.10 ^a^	0.07	0.001
C24:1 (%)	0.11 ^a^	0.59 ^b^	0.45 ^b^	0.50 ^b^	0.012	0.003
PUFA						
C18:2 *n*-6 (%)	17.67 ^b^	17.64 ^b^	18.52 ^a^	17.74 ^b^	0.34	0.05
C18:3 *n*-6 (%)	0.23	0.21	0.22	0.22	0.016	0.184
C18:3 *n*-3 (%)	0.47 ^b^	0.47 ^b^	0.53 ^a^	0.49 ^b^	0.031	0.001
C20:2 *n*-6 (%)	0.53	0.59	0.47	0.64	0.07	0.189
C20:3 *n*-6 (%)	0.66	0.63	0.54	0.60	0.04	0.091
C20:4 *n*-6 (%)	0.028 ^a^	0.024 ^b^	0.024 ^b^	0.026 ^ab^	0.0017	0.012
C20:5 *n*-3 (%)	0.21 ^b^	0.25 ^ab^	0.23 ^b^	0.27 ^a^	0.018	0.041
C22:6 *n*-3(%)	0.91 ^b^	1.19 ^a^	1.30 ^a^	1.22 ^a^	0.047	0.001
Total SFA (%)	41.16 ^a^	40.00 ^b^	39.07 ^c^	40.29 ^b^	0.25	0.001
Total MUFA (%)	32.07	32.02	32.59	30.73	0.92	0.291
Total PUFA (%)	20.72 ^b^	21.03 ^ab^	21.83 ^a^	21.18 ^ab^	0.38	0.047

^1^ Control, basic diet; CYP1, CYP2, and CYP3 represent the basal diet supplemented with 250, 500, and 1000 mg/kg CYP, respectively. ^2^ SEM, standard error of the mean. ^a,b,c^ In the same column, values with different small letter superscripts mean significant difference (*p* < 0.05).

**Table 5 foods-12-01479-t005:** Effects of CYP on fatty acids composition from thigh muscle in broilers.

Item	CYP Level (mg/kg) ^1^				SEM ^2^	*p*-Value
	Control	CYP1	CYP2	CYP3		
SFA						
C14:0 (%)	0.67 ^a^	0.60 ^a^	0.45 ^b^	0.63 ^a^	0.040	0.003
C16:0 (%)	22.02 ^a^	21.05 ^b^	21.30 ^b^	22.00 ^a^	0.28	0.018
C17:0 (%)	2.64	2.28	2.25	2.59	0.16	0.094
C18:0 (%)	5.19 ^a^	4.32 ^b^	3.84 ^b^	4.34 ^b^	0.31	0.015
C20:0 (%)	0.074 ^a^	0.046 ^b^	0.037 ^b^	0.044 ^a^	0.0060	0.001
MUFA						
C14:1(%)	0.050 ^a^	0.046 ^ab^	0.042 ^b^	0.049 ^a^	0.0030	0.05
C16:1 (%)	4.25 ^a^	4.28 ^b^	3.06 ^b^	3.11 ^a^	0.26	0.002
C18:1 (%)	39.50 ^b^	41.75 ^a^	42.30 ^a^	39.39 ^b^	0.97	0.033
C20:1 (%)	0.41 ^a^	0.41 ^a^	0.32 ^b^	0.35 ^b^	0.02	0.006
C22:1 (%)	0.10	0.0.091	0.083	0.085	0.030	0.916
C24:1 (%)	0.073 ^a^	0.079 ^a^	0.052 ^b^	0.055 ^b^	0.0040	0.001
PUFA						
C18:2 *n*-6 (%)	20.42 ^b^	20.53 ^b^	21.43 ^a^	20.91 ^ab^	0.26	0.017
C18:3 *n*-6 (%)	0.19	0.19	0.23	0.18	0.024	0.23
C18:3 *n*-3 (%)	0.77 ^a^	0.61 ^b^	0.88 ^a^	0.77 ^a^	0.052	0.001
C20:2 *n*-6 (%)	0.33	0.26	0.28	0.32	0.06	0.647
C20:3 *n*-6 (%)	0.32	0.21	0.39	0.41	0.09	0.183
C20:4 *n*-6 (%)	0.034 ^b^	0.038 ^b^	0.038 ^b^	0.048 ^a^	0.0038	0.037
C20:5 *n*-3 (%)	0.059 ^b^	0.10 ^a^	0.11 ^a^	0.11 ^a^	0.0066	0.001
C22:6 *n*-3(%)	0.49 ^b^	0.71 ^a^	0.73 ^a^	0.58 ^b^	0.043	0.001
Total SFA (%)	30.59 ^a^	28.30 ^b^	27.89 ^b^	29.60 ^a^	0.55	0.001
Total MUFA (%)	44.40 ^bc^	46.66 ^a^	45.87 ^ab^	43.04 ^c^	0.87	0.014
Total PUFA (%)	22.59 ^c^	22.59 ^c^	24.07 ^a^	23.36 ^b^	0.30	0.003

^1^ Control, basic diet; CYP1, CYP2, and CYP3 represent the basal diet supplemented with 250, 500, and 1000 mg/kg CYP, respectively. ^2^ SEM, standard error of the mean. ^a,b,c^ In the same column, values with different small letter superscripts mean significant difference (*p* < 0.05).

**Table 6 foods-12-01479-t006:** Effects of CYP on fatty acid nutrition index in breast muscle of broilers.

Item ^2^	CYP Level (mg/kg) ^1^				SEM ^3^	*p*-Value
	Control	CYP1	CYP2	CYP3		
PUFA: SFA	0.50 ^b^	0.53 ^b^	0.56 ^a^	0.53 ^b^	0.01	0.006
*n*-6: *n*-3	12.00 ^a^	9.84 ^b^	9.63 ^b^	9.86 ^b^	0.22	0.001
DHA + EPA (%)	1.13 ^b^	1.46 ^a^	1.52 ^a^	1.47 ^a^	0.04	0.001
UI	4.83	4.69	4.67	4.70	0.16	0.766
PI	0.30 ^b^	0.33 ^a^	0.34 ^a^	0.33 ^a^	0.006	0.002
NVI	1.62	1.59	1.61	1.52	0.05	0.304
IA	0.52	0.50	0.48	0.53	0.015	0.063
IT	1.20 ^a^	1.13 ^b^	1.07 ^c^	1.15 ^ab^	0.02	0.002
HHR	1.94	2.03	2.09	1.93	0.058	0.078
HPI	1.92	2.01	2.07	1.90	0.056	0.052

^1^ Control, basic diet; CYP1, CYP2, and CYP3 represent the basal diet supplemented with 250, 500, and 1000 mg/kg CYP, respectively. ^2^ DHA = docosahexaenoic acid, C22:6 n-3; EPA = eicosapentaenoic acid, C20:5 *n*-3; UI = fatty acid unsaturation index; PI = peroxidation trend index; NVI = nutrition value index; IA = index of atherogenicity; IT = index of thrombogenicity; HHR = hypocholesterolemic/hypercholesterolemic ratio; HPI = health-promoting index. ^3^ SEM, standard error of the mean. ^a,b,c^ In the same column, values with different small letter superscripts mean significant difference (*p* < 0.05).

**Table 7 foods-12-01479-t007:** Effects of CYP on fatty acid nutrition index in thigh muscle of broilers.

Item ^2^	CYP Level (mg/kg) ^1^				SEM ^3^	*p*-Value
	Control	CYP1	CYP2	CYP3		
PUFA: SFA	0.74 ^c^	0.80 ^b^	0.86 ^a^	0.79 ^b^	0.016	0.001
*n*-6: *n*-3	16.98 ^a^	14.81 ^a^	12.90 ^b^	15.10 ^a^	0.636	0.007
DHA + EPA (%)	0.55 ^c^	0.82 ^a^	0.85 ^a^	0.69 ^b^	0.046	0.001
UI	4.69 ^a^	3.76 ^b^	5.36 ^a^	5.05 ^a^	0.304	0.004
PI	0.29 ^c^	0.30 ^b^	0.33 ^a^	0.31 ^b^	0.005	0.001
NVI	2.03 ^b^	2.19 ^a^	2.17 ^a^	1.99 ^b^	0.041	0.003
IA	0.37 ^a^	0.34 ^b^	0.33 ^b^	0.37 ^a^	0.006	0.001
IT	0.76 ^a^	0.68 ^b^	0.65 ^b^	0.73 ^a^	0.016	0.001
HHR	2.73 ^b^	2.97 ^a^	3.05 ^a^	2.77 ^b^	0.047	0.001
HPI	2.71 ^b^	2.96 ^a^	3.02 ^a^	2.71 ^b^	0.047	0.001

^1^ Control, basic diet; CYP1, CYP2, and CYP3 represent the basal diet supplemented with 250, 500, and 1000 mg/kg CYP, respectively. ^2^ DHA = docosahexaenoic acid, C22:6 *n*-3; EPA = eicosapentaenoic acid, C20:5 *n*-3; UI = fatty acid unsaturation index; PI = peroxidation trend index; NVI = nutrition value index; IA = index of atherogenicity; IT = index of thrombogenicity; HHR = hypocholesterolemic/hypercholesterolemic ratio; HPI = health-promoting index. ^3^ SEM, standard error of the mean. ^a,b,c^ In the same column, values with different small letter superscripts mean significant difference (*p* < 0.05).

## Data Availability

Data supporting the results of this study can be provided by the corresponding authors on reasonable demand.

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
