# Peer review of "Effects of Chinese Yam Polysaccharide on Intramuscular Fat and Fatty Acid Composition in Breast and Thigh Muscles of Broilers"

_foods, 2023, doi:10.3390/foods12071479_

Round 1

Reviewer 1 Report

Manuscript foods-2221574, entitled “Effects of Chinese Yam Polysaccharide on intramuscular fat and fatty acid composition in breast and thigh muscles of broilers

This article provides useful information describing the effects of Chinese Yam Polysaccharide on intramuscular fat and fatty acid composition in breast and thigh muscles of broilers. Some points should be corrected or clarified. There are also a lot of grammar, stylistic and syntax errors that in some cases negatively influence the understanding of the text.

General comments:

1.      Too high IMF percentages in Table 3. In commercial hybrids, IMF content is approximately 1 and 4% in breast and thigh muscle, respectively.

2.      In abstract, the presentation of your results is not correct, since the conclusion that “dietary supplementation with 500 mg/kg CYP can up-regulate the mRNA expression levels of Wnt1 and β-catenin in breast muscle” is not completely correct. All CYP groups show the same effect. Moreover, “mRNA expression levels of PPARγ and C/EBPα in breast and thigh muscles could be down-regulated” in all CYP groups, not only in CYP2.

3.      Welfare issues exist, since the light regime of the facility was 24h light per day. Not even 1 h of dark period.

4.      In Tables 6-7, please explain abbreviations as a footnote under the Table.

5.      Discussion is inadequate, since authors did not discuss their findings and compare them with the existing literature, but they refer to the significance of each index.

Minor points

Please check the attached file

Author Response

Response to Reviewer1 Comments

Dear Reviewer,

Thank you very much for your questions. All questions from you we have answered as follows:

  1. Too high IMF percentages in Table 3. In commercial hybrids, IMF content is approximately 1 and 4% in breast and thigh muscle, respectively.

Response 1: Thank you very much for your advice. IMF is higher than 4% because we measure intramuscular fat in dried meat products. (In red: Line 124-125).

  1. In abstract, the presentation of your results is not correct, since the conclusion that “dietary supplementation with 500 mg/kg CYP can up-regulate the mRNA expression levels of Wnt1 and β-catenin in breast muscle” is not completely correct. All CYP groups show the same effect. Moreover, “mRNA expression levels of PPARγ and C/EBPα in breast and thigh muscles could be down-regulated” in all CYP groups, not only in CYP2.

Response 2: Thank you very much for your advice. We have redescribed the results. (In red: Line 186-188, 191-195).

  1. Welfare issues exist, since the light regime of the facility was 24h light per day. Not even 1 h of dark period.

Response 3: Thank you very much for your advice. Only the chicks had 24 hours of light in the first three days, and from the fourth day they had periodic light exposure.

  1. In Tables 6-7, please explain abbreviations as a footnote under the Table. 

Response 4: Thank you very much for your advice. We have annotated the abbreviations below Table 6-7.

  1. Discussion is inadequate, since authors did not discuss their findings and compare them with the existing literature, but they refer to the significance of each index.

Response 5: Thank you very much for your advice. We also added relevant contents to the discussion section. (In red: Line 334-343).

Minor points

Please check the attached file

Response : Thank you very much for your advice. We have checked the attached file.

Reviewer 2 Report

the paper aimed to investigate the effects of Chinese Yam Polysaccharide on intramuscular fat and fatty acid composition in breast and thigh muscles of broilers. The Topic is innovant and imortant, since it is considered a balanced diet to ensure the animal health and welfare, and therfore a good quality products which respond to the requirements of the consumer.

In the introduction section,  you have to add more details about the Chinese yam polysaccharide (CYP) dietary supplementation, its importance in the international context, the benefits etc.... its economic and environmental importance?

in the material and methods section, it is better to add the caracteristic and /or the composition of hinese yam polysaccharide (CYP) dietary supplementation.

Which is the correct body weight (39±1g, / 39.54 ± 0.51 g????? 

which is the average weight of bird for the sample analysis?? add a value 

- The IMF of the content of breast and thigh was measured by the Soxhlet extraction method. According to the method of Sukhija et al. to determine the composition of fatty acids [27]. ???  the sentence is not understandable 

-PUFAs: SFAs were also calculated. add the formula  or the eqquation and or the references??

for the data analysis , the multuple Duncan was used. However, in your experiment design you have mentioned that you have used a control and to compare each treatment with control , the test Dunnet mus be used. in this case you have to use it and the anva table will  allow you to compare each treatment to control one.

The results will be uabdated according to the test will be used in your  study

In your case , you have to revise the statistical model , test and therfore, the results will be affected 

In conclusion the curret results can not be consudered suitable for publication , since you dont have used the appropriate test. 

Author Response

Response to Reviewer 2 Comments

Dear Reviewer,

Thank you very much for your questions. All questions from you we have answered as follows:

In the introduction section, you have to add more details about the Chinese yam polysaccharide (CYP) dietary supplementation, its importance in the international context, the benefits etc.... its economic and environmental importance?

Response: Thank you very much for your advice. We have added the benefits of yam polysaccharide as a dietary supplement in culture in the introduction section. (In red: Line 82-86).

in the material and methods section, it is better to add the caracteristic and /or the composition of hinese yam polysaccharide (CYP) dietary supplementation.

Response: Thank you very much for your advice. We have added key components of CYP to our materials and methods (monosaccharide types include glucose 99.48% and galactose 0.52%). (In red: Line 93-94).

Which is the correct body weight (39±1g, / 39.54 ± 0.51 g?????

Response: Thank you very much for your advice. The 39±1g is the correct body weight. (In red: Line 97).

which is the average weight of bird for the sample analysis?? add a value

Response: Thank you very much for your advice.  We have added the average body weight of the sample to the materials and methods. (In red: Line 117-118).

- The IMF of the content of breast and thigh was measured by the Soxhlet extraction method. According to the method of Sukhija et al. to determine the composition of fatty acids [27]. ??? the sentence is not understandable

Response: Thank you very much for your advice. We have made changes to the language. (In red: Line 124-126).

-PUFAs: SFAs were also calculated. add the formula or the eqquation and or the references??

Response: Thank you very much for your advice. We have added relevant formulas for calculating PUFAs:SFAs to the materials and methods. (In red: Line 138).

for the data analysis , the multuple Duncan was used. However, in your experiment design you have mentioned that you have used a control and to compare each treatment with control , the test Dunnet mus be used. in this case you have to use it and the anva table will allow you to compare each treatment to control one. 

Response: Thank you very much for your advice.  As the purpose of our study was to screen out the optimal addition amount of yam polysaccharide, it was necessary to compare not only between the experimental group and the control group, but also between experimental groups. Therefore, Duncan multiple comparison was used for verification.

The results will be uabdated according to the test will be used in your study

In your case , you have to revise the statistical model , test and therfore, the results will be affected

In conclusion the curret results can not be consudered suitable for publication , since you dont have used the appropriate test.

Reviewer 3 Report

The purpose of this study was to determine the effect of Chinese yam polysaccharide on intramuscular  fatty acid composition in lipid breast and thigh muscles of broiler chickens. The Introduction chapter provides an overview of the world's knowledge on this subject. The material used in the research is sufficiently numerous, but some supplementing the description in Materials and Methods chapter are needed. The results are described usually correctly. The discussion is exhaustive. Summary of the results are correct. Some corrections are needed. The proposed changes are listed below.

General comments:

Please, prepare the article in accordance with the instructions for the authors:

For significance, use a low letter "p" (p < 0.05) in italic instead of the capital "P" in the main article

Tables 4 & 5 use Palatino Linotype 10 font, 1 line spacing, no line spacing (0)

In Reference chapter for page ranges use long (-) from the symbol function, instead of short (-) from the keyboard

Detailed Comments:

Page 2

Materials and Methods, subchapter 2.1. Line 6, give the commercial name of the broilers used in the experiment, L9 29-48 d? or 28-48 d?

Page 3

Provide information on relative humidity (range), color, length, light intensity, type of light incandescent? or fluorescent?

Section 2.2. L2 add (Total 24 birds) after „treatment group”?

Section 2.3. Line 2 Sukhija et al. [27], L3 delete [27]

Chapter 2.4.

Page 4

respectively? There are 7 indexes and only 5 references

In the chapter Refrences there is no “Abrami et al. "

I suggest adding a reference number after each author, for example Logue et al. […], Witting et al. […]

Page 5, subchapter 3.2., l4, „highe” or "Lower"?

Figure 1A Wnt1, not Wnt

I suggest entering the notations A, B, C, D for Figures 1 and 2

Page 6

Subchapter 3.3, Line 2-3 "...and the CYP2 group was significantly lower than CYP1, CYP3 and control group" insted of current form

L23-24 CYP1 groups, the broilers in the CYP2 and CYP3… no significance between CYP1 and CYP2 for MUFA concentrations

Table 4 and 5 - see General comments

Page 8

Line 6 below the table, CYP1 and CYP3 instead of CYP1 and CYP2

L9 under the table "higher" instead of "lower", Line 12 "higher" instead of "lower", L18 CYP 3 and control group for C22:6 not significant

Why is the Sum of SFA + MUFA + PUFA different from 100%?

Page 10

Section 3.4. Line 10 - CYP1 and CYP3 instead of current form

Page 11

Table 7 DHA+EPA instead of current form, standardization

In References chapter,

Items 15, 17, 20, 22, 24, 42 - correction of the title of the journal required.

Author Response

Response to Reviewer 3 Comments

Dear Reviewer,

Thank you very much for your questions. All questions from you we have answered as follows:

General comments:

Please, prepare the article in accordance with the instructions for the authors:

For significance, use a low letter "p" (p < 0.05) in italic instead of the capital "P" in the main article

Tables 4 & 5 use Palatino Linotype 10 font, 1 line spacing, no line spacing (0)

In Reference chapter for page ranges use long (-) from the symbol function, instead of short (-) from the keyboard

 Response: Thank you very much for your advice. We have amended the issue of appeal.

Detailed Comments:

Page 2

Materials and Methods, subchapter 2.1. Line 6, give the commercial name of the broilers used in the experiment, L9 29-48 d? or 28-48 d?

 Response: Thank you very much for your advice. Because 28 days involves sampling and requires fasting for 24 hours, so the meaning of 28 days dividing line. We have changed it to 29-48d in the paper. (In red: Line 100)

Page 3

Provide information on relative humidity (range), color, length, light intensity, type of light incandescent? or fluorescent?

 Response: Thank you very much for your advice. We have added relative humidity and lighting types to the paper. (In red: Line 107-109)

Section 2.2. L2 add (Total 24 birds) after „treatment group”?

 Response: Thank you very much for your advice.  We have added total 24 birds) after “treatment group”. (In red: Line 117)

Section 2.3. Line 2 Sukhija et al. [27], L3 delete [27]

 Response: Thank you very much for your advice. We have put [] behind Sukhija et al. (In red: Line 126)

Chapter 2.4.

Page 4

respectively? There are 7 indexes and only 5 references

 Response: Thank you very much for your advice. We measured seven indicators but had five references because IT and IA referenced the same reference, NVI and HPI referenced the same reference.

In the chapter Refrences there is no “Abrami et al. "

 Response: Thank you very much for your advice. We have checked the references and removed Abrami et al.

I suggest adding a reference number after each author, for example Logue et al. […], Witting et al. […]

 Response: Thank you very much for your advice. We have put [] behind each author.  (In red: Line 136, 137)

Page 5, subchapter 3.2., l4, “highe” or "Lower"?

 Response: Thank you very much for your advice. It is “lower”, we have changed that.  (In red: Line 185)

Figure 1A Wnt1, not Wnt

 Response: Thank you very much for your advice. We have changed Wnt to Wnt1.

I suggest entering the notations A, B, C, D for Figures 1 and 2

 Response: Thank you very much for your advice. We have entered A, B, C, D into Figures 1 and 2.

Page 6

Subchapter 3.3, Line 2-3 "...and the CYP2 group was significantly lower than CYP1, CYP3 and control group" insted of current form

 Response: Thank you very much for your advice. We have amended the issue of appeal.  (In red: Line 211)

L23-24 CYP1 groups, the broilers in the CYP2 and CYP3… no significance between CYP1 and CYP2 for MUFA concentrations

 Response: Thank you very much for your advice. We have amended the issue of appeal.  (In red: Line 231-233)

Table 4 and 5 - see General comments

 Response: Thank you very much for your advice. We have adjusted the format of Tables 4 and 5.

Page 8

Line 6 below the table, CYP1 and CYP3 instead of CYP1 and CYP2

 Response: Thank you very much for your advice. We have used CYP1 and CYP3 instead of CYP1 and CYP2  (In red: Line 252)

L9 under the table "higher" instead of "lower", Line 12 "higher" instead of "lower", L18 CYP 3 and control group for C22:6 not significant

 Response: Thank you very much for your advice. We have amended the issue of appeal.  (In red: Line 255)

Why is the Sum of SFA + MUFA + PUFA different from 100%?

 Response: Thank you very much for your advice. The sum of SFA + MUFA + PUFA is different from 100 because there are 20 FAs measured, 19 of which were analyzed.

Page 10

Section 3.4. Line 10 - CYP1 and CYP3 instead of current form

 Response: Thank you very much for your advice. We have replaced the current form with CYP1 and CYP3.  (In red: Line 287)

Page 11

Table 7 DHA+EPA instead of current form, standardization

  Response: Thank you very much for your advice. We have standardized DHA+EPA.

In References chapter,

Items 15, 17, 20, 22, 24, 42 - correction of the title of the journal required.

 Response: Thank you very much for your advice. We have made corrections to items 15, 17, 20, 22, 24, 42 - Journal title.

Round 2

Reviewer 1 Report

Authors did not include any of my corrections and comments, so I think that the article is not appropriate for publication in its present form

Author Response

Dear Reviewer,

Thank you very much for your questions. We are very sorry that we did not check the content in your attachment due to our negligence. I'm sorry for the delay. We have revised the language according to your comments and explained your comments. Due to the change of the manuscript version, we have revised it in the new version of "foods-2221574-cover letter".

Sincerely,

Zhiguo Miao
